# Identification and Mapping of QTLs for Adult Plant Resistance in Wheat Line XK502

**DOI:** 10.3390/plants13172365

**Published:** 2024-08-25

**Authors:** Xianli Feng, Ming Huang, Xiaoqin Lou, Xue Yang, Boxun Yu, Kebing Huang, Suizhuang Yang

**Affiliations:** Wheat Research Institute, School of Life Sciences and Engineering, Southwest University of Science and Technology, Mianyang 621010, China; 13882742747@163.com (X.F.); 18228961047@163.com (M.H.); 15681911926@163.com (X.L.); 19166901428@163.com (X.Y.); yuboxunhj@163.com (B.Y.); huangkebing@swust.ede.cn (K.H.)

**Keywords:** wheat stripe rust, adult plant resistance, quantitative trait loci, single-nucleotide polymorphism

## Abstract

Stripe rust is a serious wheat disease occurring worldwide. At present, the most effective way to control it is to grow resistant cultivars. In this study, a population of 221 recombinant inbred lines (RILs) derived via single-seed descent from a hybrid of a susceptible wheat line, SY95-71, and a resistant line, XK502, was tested in three crop seasons from 2022 to 2024 in five environments. A genetic linkage map was constructed using 12,577 single-nucleotide polymorphisms (SNPs). Based on the phenotypic data of infection severity and the linkage map, five quantitative trait loci (QTL) for adult plant resistance (APR) were detected using the inclusive composite interval mapping (ICIM) method. These five loci are *QYrxk502.swust-1BL*, *QYrxk502*.*swust*-*2BL*, *QYrxk502*.*swust*-*3AS*, *QYrxk502*.*swust*-*3BS*, and *QYrxk502*.*swust*-*7BS*, explaining 5.67–19.64%, 9.63–36.74%, 9.58–11.30%, 9.76–23.98%, and 8.02–12.41% of the phenotypic variation, respectively. All these QTL originated from the resistant parent XK502. By comparison with the locations of known stripe rust resistance genes, three of the detected QTL, *QYrxk502*.*swust*-3AS, *QYrxk502*.*swust*-*3BS*, and *QYrxk502*.*swust*-*7BS*, may harbor new, unidentified genes. From among the tested RILs, 16 lines were selected with good field stripe rust resistance and acceptable agronomic traits for inclusion in breeding programs.

## 1. Introduction

Wheat stripe rust, caused by *Puccinia striiformis* Westend. f. sp. *tritici* Erikss. (*Pst*) [1], is one of the most serious diseases in the world. Losses due to stripe rust typically range from 10% to 70% in commercial production environments [2]. Between 1975 and 2012, the average losses of susceptible cultivars of winter and spring wheat near Pullman, Washington, USA, were 36% and 30%, respectively [3]. In 2008, wheat stripe rust in Punjab, India, caused losses of about INR 236 million [4]. In 2010, wheat stripe rust that swept through Central and West Asia caused yield losses of 20% to 70% in different countries [5]. In the winter and spring wheat regions of Northwest China, Southwest China, North China, and the Yellow and Huaihai Seas, the annual incidence of stripe rust is estimated to be about 4–5.3 million hectares [6].

The identification and deployment of new genes for stripe rust resistance is an ongoing struggle [7,8,9,10]. To date, 86 stripe rust resistance genes (*Yr1*–*Yr86*) have been formally named, mainly from common wheat [11]. In addition, more than 300 quantitative trait loci (QTL) have been reported [11,12,13,14,15]. According to the expression period of stripe rust resistance genes, they can be divided into all-stage resistance (ASR) genes and adult plant resistance (APR) genes. A single ASR gene is easily overcome by new *Pst* races [5,16,17,18]. Adult plant resistance genes usually provide race-nonspecific resistance and are more durable, which can better solve the current difficulties faced by wheat stripe rust resistance breeding [19,20]. Combining APR genes with effective ASR genes in the future is the best way to breed wheat cultivars with high levels of durable resistance [21].

At present, single-nucleotide polymorphism (SNP) markers are often used for quantitative trait locus (QTL) detection, which has higher accuracy and density than other molecular markers [22,23,24,25,26,27,28]. It has been widely used in wheat QTL positioning and whole-genome association analysis, providing valuable markers and information for genetic analysis and breeding [29]. The wheat 55K SNP array was developed by the Chinese Academy of Agricultural Sciences based on the 660K SNP array combined with thousands of local materials [30]. It is more suitable for the research of domestic wheat germplasm materials and is also of great significance for cultivar identification and gene positioning [31].

In this study, 221 RILs and a 55K SNP array consisting of the mapping population SY95-71/XK502 were used to profile the wheat line XK502, which showed a high level of resistance in many years of field trials, to explore the stripe rust resistance line of XK502, detect QTLs, obtain molecular markers closely linked to them, and identify QTLs by comparing their chromosomal locations with previously reported stripe rust resistance QTLs.

## 2. Results

### 2.1. Phenotypic Analysis

According to the seedling identification in the greenhouse, the parents SY95-71, XK502, and the susceptible control Mingxian 169 (MX169) showed high susceptibility with infection type (IT) = 8,9 to *Pst* minor races CRY32, CRY33, and CRY34 (Figure 1A–C), and the surface of the leaves was covered with numerous spore mounds.

The identification of wheat stripe rust resistance took place in five environments: in Jiangyou (JY; 31°31′ N, 104°51′ E) in 2022 (22), in both Mianyang (MY; 31°27′ N, 104°68′ E) and Jiangyou in 2023 (23), and in the experimental fields of Guangyuan (GY; 31°88′ N, 106°01′ E) and Jiangyou in 2024 (24), in Sichuan Province. In all environments, SY95-71 showed high susceptibility, with IT = 8,9 and a disease severity (DS) of ≥85% (Figure 1D); the MX169 was also highly susceptible (IT = 8,9; DS ≥ 80%). The resistant parent XK502 (Figure 1D) exhibited high resistance characteristics (IT = 1–3, DS = 0–5%), suggesting that XK502 is a resistant line at the adult plant stage. The IT of the RIL population was in the range of 0–9 (Figure 2), and DS was in the range of 0–100%; additionally, IT and DS were continuously distributed and approximately normally distributed, indicating the presence of quantitative trait loci in the SY95-71/XK502 recombinant inbred line population (Figure 3).

A correlation analysis showed that 221 RILs had significant correlations between IT and DS in the five environments (r = 0.49–0.75, *p* < 0.001 for IT; r = 0.53–0.73, *p* < 0.001 for DS) (Table 1). In the analysis of variance (ANOVA), IT and DS showed extremely significant differences among different genotypes, different environments, and the interaction between different genotypes × different environments (*p* < 0.001). The broad-sense heritability (h^2^_b_) of IT was 0.90 and that of DS was 0.89, indicating that this trait variation was less affected by the environment and was mainly controlled by genes. Resistance genes play an important role in reducing the severity of the disease (Table 2).

### 2.2. QTL Analysis of Stripe Rust Resistance

A genetic map was constructed using 12,577 SNP markers with known chromosomal locations. On average, there were 599 markers distributed on each chromosome, and the average genetic distance between two markers was 0.97 cM. The genetic map was combined with IT and DS data to preliminarily detect the stripe rust resistance QTL using the inclusive composite interval mapping (ICIM) method. A total of five adult plant resistance QTLs were detected, which were located at chromosomes 1BL, 2BL, 3AS, 3BS, and 7BS, tentatively named *QYrxk502.swust-1BL*, *QYrxk502.swust-2BL*, *QYrxk502.swust-3AS*, *QYrxk502.swust-3BS*, and *QYrxk502.swust-7BS*, respectively. All QTLs were derived from the resistant parent XK502 (Appendix A, Figure 4). Among them, *QYrxk502.swust-1BL*, *QYrxk502.swust-2BL*, *QYrxk502.swust-3BS*, and *QYrxk502.swust-7BS* have appeared in four or more environments and were considered stable QTLs.

*QYrxk502.swust-1BL*, located between markers *AX-109335890* and *AX-109389405*, with a genetic distance of 80.46–81.54 cM, explained 5.96–19.64% and 5.67–19.60% of the phenotypic variation of IT and DS, respectively. *QYrxk502.swust-2BL*, located between markers *AX-108884194* and *AX-110024591*, with a genetic distance of 361.45–362.20 cM, explained 10.04–36.74% and 9.63–34.50% of the phenotypic variation of IT and DS, respectively. *QYrxk502.swust-3AS*, located between markers *AX-111631905* and *AX-109308178*, with a genetic distance of 307.78–317.30 cM, explained 10.26–10.88% and 9.58–11.30% of the phenotypic variation of IT and DS, respectively. *QYrxk502.swust-3BS*, located between markers *AX-108747357* and *AX-109438796*, with a genetic distance of 47.20–57.27 cM, explained 9.76–20.79% and 11.27–23.98% of the phenotypic variation of IT and DS, respectively. *QYrxk502.swust-7BS*, located between markers *AX-109968088* and *AX-110982135*, with a genetic distance of 405.25–406.79 cM, explained 8.02–11.90% and 8.81–12.41% of the phenotypic variation of IT and DS, respectively. Different effects of the same QTL in different environments may be caused by specific environments and different disease pressures. Inclusive composite interval mapping of digenic EPI static (ICIM-EPI) analysis was used to perform pairwise analysis of the QTL regions of chromosomes 1BL, 2BL, 3AS, 3BS, and 7BS. The results showed that there was no epistasis between these five QTLs.

Among the five QTLs, *QYrxk502.swust-1BL* and *QYrxk502*.*swust*-*3BS* overlapped with *Yr29* and *Yr30* in Chinese spring wheat and had the same resistance type, so these two QTLs may be *Yr29* and *Yr30*. However, *Yr29* is closely linked to the leaf tip necrosis (*LTN*) gene (leaf tip necrosis occurs when this gene is present), but XK502 was not observed to exhibit leaf tip necrosis in this study, so *QYrxk502*.*swust*-*1BL* was not *Yr29*. *Yr30* was linked to the morphological marker pseudo-black husk (PBC), and wheat carrying this gene will gradually turn black in the husk and internodes in the late grain-filling period. Although the resistant parent XK502 in this study showed the characteristic of blackening of glumes in the late grain-filling period (Figure 5A), we detected the flanking marker *WMS533* of *Yr30* but failed to find it in XK502 (Figure 5B), so *QYrxk502*.*swust*-*3BS* was not *Yr30* either.

### 2.3. Additive Effect Analysis for QTL

In order to determine the effects of different QTL on stripe rust, 221 RILs were divided into five groups (Table 3), and the effect sizes of these five QTL were determined based on the average infection type and disease severity as *QYrxk502.swust-3BS* > *QYrxk502*.*swust*-*2BL* > *QYrxk502*.*swust*-*7BS* > *QYrxk502*.*swust*-*3AS* > *QYrxk502*.*swust*-*1BL*. Obviously, the average IT and DS of RILs carrying any QTL were lower than those without QTL. The 221 recombinant inbred lines were divided into six groups according to the number of QTLs they contained (Figure 6). The average IT and DS of RILs without QTL were 8.21 and 82.57%, respectively, which were similar to those of the susceptible parent SY95-71. The average IT and DS of RILs containing one QTL were 7.12 and 70.60%, respectively, which were 13.42% and 14.50% lower than those without QTLs; the average IT and DS of RILs containing two QTL were 5.04 and 40.97%, respectively, which were 38.61% and 50.38% lower than those without QTL; the average IT and DS of RILs containing three QTL were 4.39 and 30.60%, respectively, which were 46.53% and 62.94% lower than those without QTL. The average IT and DS of RILs containing four QTL were 3.46 and 17.55%, respectively, which were 57.86% and 78.75% lower than those without QTL. The average IT and DS of RILs containing five QTL were 2.9 and 7.8%, respectively, which were reduced by 64.68% and 90.55% compared with RILs without QTL, and were comparable to the resistant parent XK502. This indicates that the clustering of multiple QTL for adult plant resistance can enhance wheat resistance to stripe rust.

### 2.4. Selection of Breeding Lines

The average plant height (PH) of the parents SY95-71 and XK502 was 83.58 cm and 97.75 cm, respectively, and the average PH of the RILs was 66.92–110.83 cm. The average productive tiller number (PTN) of SY95-71 and XK502 was 7 and 9, respectively, and the average PTN of the RILs was 5–12. The average spike length (SL) of SY95-71 and XK502 was 8.33 cm and 10.38 cm, respectively, and the average SL of RILs was 6.68–12.83 cm. The average thousand-kernel weight (TKW) of SY95-71 and XK502 was 33.68 g and 44.12 g, respectively, and the average TKW of the RILs was 28.78–61.14 g. The average grain length (GL) of SY95-71 and XK502 was 5.5 and 6.62 mm, respectively, and the average GL of the RILs was 5.29–7.08 mm. The average grain width (GW) of SY95-71 and XK502 was 2.92 and 3.14 mm, respectively, and the average GW of the RILs was 2.53–3.65 mm. The average grain length–width ratio (LWR) of SY95-71 and XK502 was 1.90 and 2.12, respectively, and the average LWR of the RILs was 1.71–2.34. Correlation analysis showed that IT and DS were extremely significantly negatively correlated with the first six agronomic traits (*p* < 0.001), which showed that the key to improving wheat agronomic traits is to ensure the resistance of the crop (Table 4).

The criteria for screening RILs are as follows: PH = 80–90 cm, PTN = 6–11 spikes, spike length > 8.5 cm, TKW > 40 g, grain length > 5.5 mm, grain width > 3 mm, length–width ratio = 1.5–2.5. A total of 16 eligible families were selected from the 221 RILs (Appendix A). These lines contained at least two QTL and, at most, four QTL.

## 3. Discussion

Genes for adult plant resistance are important in wheat stripe rust resistance research. They have a longer-lasting resistance than the ASR gene. In the recombinant inbred line population composed of SY95-71/XK502, we detected five adult plant resistance QTL, temporarily named *QYrxk502.swust*-*1BL*, *QYrxk502*.*swust*-*2BL*, *QYrxk502*.*swust*-*3AS*, *QYrxk502*.*swust*-*3BS*, and *QYrxk502*.*swust*-*7BS*. To determine the relationship between the QTL mapped in this study and the reported “*Yr*” genes/QTLs, we compared the relative physical distances of these loci based on Chinese spring wheat (IWGSC Ref Seq v.1.0).

*QYrxk502*.*swust*-*1BL* is located between markers *AX*-*109335890* and *AX*-*109389405* in the physical interval 670,382,321–670,593,327 bp. The genes that have been officially named on wheat chromosome 1BL are *Yr21*, *Yr26*, and *Yr29*, of which, *Yr21* [33] and *Yr26* [34] are ASR genes. *Yr29* is an APR gene that is closely linked to the SSR marker *Xwmc44* and has a physical location of 678,736,681 bp [35]. It is also closely associated with the leaf tip necrosis *LTN* gene, but no leaf tip necrosis was observed in XK502 in the field in this study, so *QYrxk502*.*swust*-*1BL* is not *Yr29*. The *Yr29* is widely distributed in CIMMYT wheat germplasm. *QYr*.*hebau*-*1BL* [36], *QYr*.*ucw*-*1BL* [37], *QYr*.*crc*-*1BL* [38], *QYr*.*cim*-*1BL* [39], *QYrCW357*-*1BL* [40], *QYr*.*sicau*-*1BL* [26], and *QYr*.*hazu*-*1BL* [41] are all likely to be *Yr29*. In addition, several tentatively named QTL have been previously reported on chromosome 1BL, with *YrNS*-1 flanking markers *Xgwm124* and *Xwmc719*, and a physical interval of 638,898,514–664,520,756 bp [42]. *QYrsv*.*swust*-*1BL*.*1* is flanked by markers *IWB5732* and *IWB4839*, with a physical range of 670,783,574–671,505,487 bp [43]. The flanking markers for *QYr*.*sdau*-*1BL* are *KASP*_*63005* and *KASP*_*19405*, with a physical interval of 655.7–677.3 Mb [9]. *QYr*.*gaas.1B*.*1* was flanked by *AX*-*108745931* and *AX*-*110017315*, with a physical interval of 667,012,868–667,229,049 bp [44]. *QYr*.*sicau*-*1B.1* is linked to SSR marker *Xwmc156*, with a physical location of 462,460,534 bp [45]. *QYrxk502*.*swust*-*1BL* lies within the physical interval of *QYr*.*sdau*-*1BL*, and further research is needed to confirm the relationship.

*QYrxk502*.*swust*-*2BL* is located between markers *AX*-*108884194* and *AX*-*110024591*, with a physical range of 683,285,668–690,205,985 bp. The genes that have been officially named on wheat chromosome 2BL are *Yr5* [46], *Yr7* [47], *Yr43* [48], *Yr44* [49], and *Yr53* [50], all of which belong to the ASR genes, whereas *QYrxk502*.*swust*-*2BL* belongs to the APR genes. In addition, several QTL were located on chromosome 2BL: *QYr*.*hbaas*-*2BL* flanking marker is *IWA586*, and its physical location is 453.3 Mb [51]. The *QYr*.*niab*-*2B*.*1* flanking marker is *Kukri*_*c9118*_*1774*, and its physical location is 683.05 Mb [52]. The *QYrPI660122*.*swust*-*2BL* flanking markers are *AX*-*109349804* and *AX*-*109849173*, with a physical interval of 777,831,275–779,847,527 bp [53]. The *QYr*.*niab*-*2B* flanking marker is *WPT*-*9190*, and its physical location is 750.12 Mb [54]. *QYrhm*.*nwafu*-*2BC* is located between markers *IWB26631* and *IWB40714* in the physical interval 212.58–215.2 Mb [26]. The *QYR1* flanking marker is *Xgwm501*, and its physical location is 672.08 Mb [55]. The *QYr*.*caas*-*2BL* flanking markers are *Xwmc441* and *Xwmc361*, with a physical interval of 598,064,477–779,339,263 bp [56]. *QYrxk502*.*swust*-*2BL* is different from *QYraq*.*cau*-*2BL* (HTAP gene), but its relationship with *QYr*.*niab*-*2B*.*1* and *QYr*.*caas*-*2BL* needs further study and confirmation.

*QYrxk502*.*swust*-*3AS* is located between markers *AX*-*109308178* and *AX*-*111631905*, with a physical range of 37,303,271–46,005,047 bp. The only officially named gene on wheat chromosome 3AS is *Yr76* [57], which is an ASR gene. There are also a number of QTL that have been located to chromosome 3AS. The *QYrsv*.*swust*-*3AS* flanking markers are *IWB7237* and *IWB8523*, with a physical interval of 21.26–22.48 Mb [43]. The *QYr*.*niab*-*3A*.*1* flanking marker is *Kukri*_*c28650*_*111*, and its physical location is 7.92 Mb [52]. The *QYr*.*hbau*-*3AS* flanking markers are *AX*-*111491666* and *AX*-*110551014*, with a physical interval of 45.78–54.87 Mb [36]. The *QYrto*.*swust*-*3AS* flanking markers are *AX*-*95240191* and *AX*-*94828890*, with a physical range of 7,920,927–10,144,518 bp [58]. The *QYr*.*spa*-*3A*.*1* flanking marker is *BS00021981*_51, and its physical location is 61,349,059 bp [59]. The *QYr*.*dms*-*3A* flanking markers are *Tdurum*_*contig74920*_*757* and *RAC875*_*c45016*_*79*, with a physical interval of 51,177,286–53,837,108 bp [60]. The *QYr*.*ifa*-*3AS* flanking markers are *wPt*-*9634* and *wPt*-*0714*, with a physical interval of 11,379,461–25,903,821 bp [61]. The *QYr*.*caas*-*3AS* flanking marker is *Kukri*_*c96747*_*274*, and its physical location is 19,181,736 bp [62]. *QYrxk502*.*swust*-*3AS* is different from these QTL and may be a new QTL.

*QYrxk502*.*swust*-*3BS* is located between markers *AX*-*109438796* and *AX*-*108747357*, with a physical distance range of 933,501–7,529,669 bp. The genes that have been officially named on wheat chromosome 3BS are *Yr4a* [11], *Yr4b* [63], *Yr30*, *Yr57* [64], and *Yr58*, among which, *Yr4a*, *Yr4b*, and *Yr57* are ASR genes, and *Yr30* and *Yr58* are APR genes. The *Yr58* flanking marker is *Xbarc75*, with a physical location of 3,395,500 bp [65]. The flanking markers of *Yr30* are *Xgwm389* and *Xgwm533*, with a physical location of 806,388–35,326,710 bp; it is linked to the morphological marker pseudo-black chaff (PBC), and if wheat contains this gene, the husk and internodes will gradually darken in the late grain-filling period [66]. The resistant parent XK502 in this study showed the trait of blackening of glumes in the late grain-filling period, and *QYrxk502*.*swust*-*3BS* was within the physical interval of *Yr30*. Therefore, we tested the chain marker *WMS533* for *Yr30* in XK502 but did not detect the marker, and *QYrxk502*.*swust*-*3BS* was not *Yr30*. *Yr30* has been mapped in many wheat germplasms. *QYrlov*.*nwafu*-*3BS* located in wheat P10057 is *Yr30*, and the flanking markers are *IWB57990* and *IWB6491*, which are linked to the flanking markers *Xgwm389* and *Xgwm533* of *Yr30*, respectively. The parent P10057 showed dark glumes in the late grain-filling period, and the color of the glumes’ offspring was basically consistent with the haplotype of the KASP marker [67]. The flanking markers of *QYr*.*hbaas*-*3BS* [51], *QYr*.*nafu*-*3BS* [68], *QYr*.*ccsu*-*3B.1* [69], *QYrto*.*swust*-*3BS* [58], *QYr*.*cim*-*3BS* [39], *QYrhm*.*nwafu*-*3BS* [70], and *QYr*.*ifa*-*3BS* [61] are within the physical interval corresponding to *Yr30* and may be *Yr30*. *QYrxk502*.*swust*-*3BS* is likely a new QTL.

*QYrxk502*.*swust*-*7BS* is located between markers *AX*-*110982135* and *AX*-*109968088*, with a physical interval of 15,508,999–18,217,648 bp. The officially named stripe rust resistance genes on wheat chromosome 7BS are *Yr6* [71] and *Yr63* [11], which are ASR genes. *QYrxk502*.*swust*-*7BS* is different from these officially named genes. Few QTLs have been reported on the short arm of chromosome 7BS, and the *QYrcw*.*nwafu*-*7BS* flanking markers AX-94670534 and AX-94488627, with physical intervals ranging from 23,490,588 to 49,139,449 bp [38], are different from *QYrxk502*.*swust*-*7BS* and are likely to be a new QTL.

## 4. Materials and Methods

### 4.1. Plant Materials

Wheat line XK502 was developed by the Wheat Research Institute of Southwest University of Science and Technology using a cross-breeding method in 2005. The line had demonstrated excellent stripe rust resistance at the adult plant stage for the last two decades in field. SY95-71 is a wheat material highly susceptible to stripe rust and is suitable for resistant breeding research [45]. SY95-71 and XK502 were hybridization, and a mapping population of 221 RILs of the F_7_ generation was obtained via single-seed descent, which was used to locate QTLs for resistance to stripe rust. MX169 and Avocet S (AvS) are both wheat cultivars that are highly susceptible to stripe rust and are often used as susceptible controls.

### 4.2. Phenotypic Identification

#### 4.2.1. Greenhouse Tests

Through a greenhouse seedling experiment, the response of SY95-71 and XK502 to stripe rust at the seedling stage was evaluated. Plants SY95-71, XK502, and MX169 were placed in small separate flowerpots with a diameter of 8 cm. Three Chinese *Pst* races, CYR32, CYR33, and CYR34, propagated in single-spore isolation were used for single-race infection. During the one-tip–one-leaf stage of wheat, the leaves were dewaxed, and then *Pst* races and talcum powder were mixed in a ratio of 1:20 and dipped in a cotton swab to apply to the front of the leaves. After the inoculation was completed, the wheat seedlings were placed in a dark environment at 8–12 °C and sprayed with water to moisturize. They were taken out after 24 h and moved to a greenhouse at 13–17 °C to continue infection. During this period, the environment was maintained with light for 16 h, no light for 8 h, and a relatively humid atmosphere. The infection type (IT) was recorded 15–20 days after inoculation using a 0–9 scale [72].

#### 4.2.2. Field Tests

For the identification of field phenotypic responses to stripe rust of 221 recombinant inbred lines from parents SY95-71, XK502, and F_6_, F_7_, and F_8_ generations, the study was conducted in 22JY, in both 23MY and 23JY, and in the experimental fields of 24GY and 24JY, in Sichuan Province. Each field trial site adopted a completely randomized block design with two replicate groups. In the experimental field, the row length was set to 1 m, and approximately 30 seeds were sown in each row, with a row spacing of 30 cm. MX169 was planted every 20 rows as a susceptible control and spore spreader to increase stripe rust pressure and uniformity in the nursery spend. The highly susceptible stripe rust cultivar AvS was planted between the experimental plots to increase the *Pst* inoculum amount. The survey method of IT was the same as that of seedling identification. The DS survey was identified and recorded according to the Cobb’s scale standard [73]. A total of three surveys were conducted, and the final results were averaged. This systematic experimental design aims to comprehensively assess the resistance of both parental and recombinant inbred line populations across various environments.

#### 4.2.3. Determination of Agronomic Traits

The agronomic traits of the parent lines SY95-71 and XK502, as well as 221 RILs, were evaluated across four environments: 23MY, 23JY, 24GY, and 24JY. We investigated PH, PTN, SL, TKW, GL, GW, and LWR [74]. By combining these seven agronomic traits, RILs with strong resistance and excellent agronomic traits were screened.

### 4.3. DNA Extraction and Genotyping

Genomic DNA was extracted from the leaves of uninfected seedlings of 221 recombinant inbred lines in the parental and F_7_ generations using a modified CTAB method [75] and the DNA concentration was diluted to 100 ng/μL in a volume of 80 μL according to the requirements of China Golden Marker (Beijing, China) Biotech. The company was commissioned to use the Affymetrix wheat 55K SNP array [76] to perform sequencing on the parental and recombinant inbred lines to obtain genotype data.

### 4.4. Genetic Linkage Map Construction and QTL Analysis

Creating a wheat genetic map is one of the key steps in conducting genome analysis and studying phenotypic variation in wheat. Previously, analysis of variance was performed on the phenotypic data of stripe rust in multiple environments to determine genetic, environmental, and genetic × environmental interaction effects, and calculate the Pearson correlation coefficient between phenotypic data in different environments [77,78]. To compare paired RIL phenotypic responses in different environments, the genetic linkage group of the recombinant inbred line was constructed through the software IciMapping4.2, and the BIN tool was first used to remove redundant markers with a deletion rate greater than 20%. Recombination rates were converted to genetic distances (cM) using the Kosambi [79] function. Genetic maps and inclusion composite interval mapping were used to detect IT and DS data of stripe rust in different environments for preliminary QTL positioning. Subsequently, in order to determine the additive effect of QTL, the effect of QTL combination was verified by plotting a box plot of the average IT and average DS of recombinant inbred lines with the same number of beneficial alleles.

## 5. Conclusions

This study successfully integrated high throughput 55K SNP sequencing with stripe rust phenotypes from both parental lines and 221 mapping populations, leading to the identification of five adult plant resistance QTL. Each of these QTL offers varying levels of resistance to stripe rust, highlighting their potential utility in crop improvement. While the associations between *QYrxk502.swust-1BL* and *QYrxk502*.*swust*-*2BL* and other QTL require further validation, *QYrxk502*.*swust*-*3AS*, *QYrxk502*.*swust*-*3BS*, and *QYrxk502*.*swust*-*7BS* appear to represent novel contributions to our understanding of stripe rust resistance. Notably, the QTL located on chromosomes 1BL, 2BL, 3BS, and 7BS exhibited stability across conditions, suggesting their reliability for breeding purposes. This study underscores the correlation between the clustering of resistance QTL and enhanced resistance levels, indicating that selecting lines with multiple QTL could yield more resilient cultivars. Furthermore, the identification of 16 promising lines from the 221 RILs through a dual approach of resistance and agronomic trait evaluation indicates a pathway toward the development of commercially viable wheat cultivars. Future efforts can focus on converting the SNP markers linked to these stable major-effect QTL into KASP or SSR markers to facilitate the breeding of wheat cultivars that possess lasting resistance to stripe rust.

## Figures and Tables

**Figure 1 plants-13-02365-f001:**
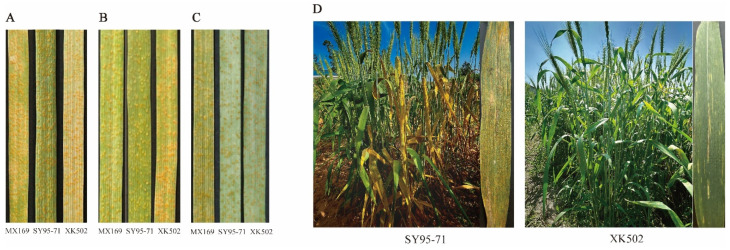
Response of the parents SY95-71 and XK502 to stripe rust at seedling and adult stages: seedlings were inoculated with CYR32 (**A**), CYR33 (**B**), and CYR34 (**C**) *Puccinia striiformis* f. sp. *tritici*; disease status of flag leaves at adult plant stage (**D**).

**Figure 2 plants-13-02365-f002:**
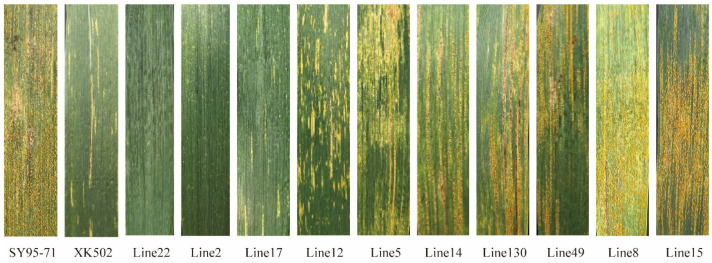
Flag leaf infection type (IT) of parental and some RIL adult plants in Jiangyou, Sichuan Province, 2024. The infection type (IT) of RILs was 0–9 from left to right.

**Figure 3 plants-13-02365-f003:**
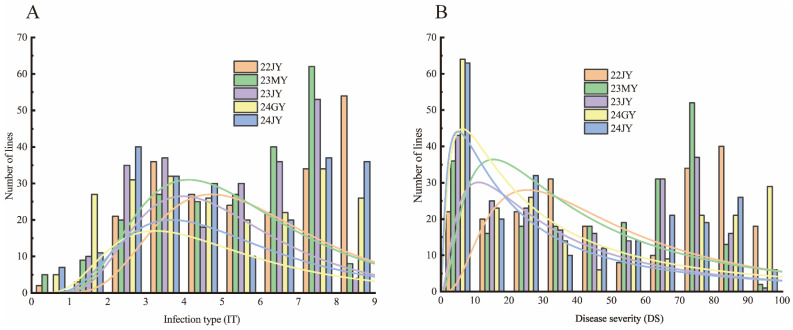
Frequency distribution diagram of average infection type (**A**) and disease severity (**B**) of 221 RILs composed of SY95-71/XK502 in five environments.

**Figure 4 plants-13-02365-f004:**
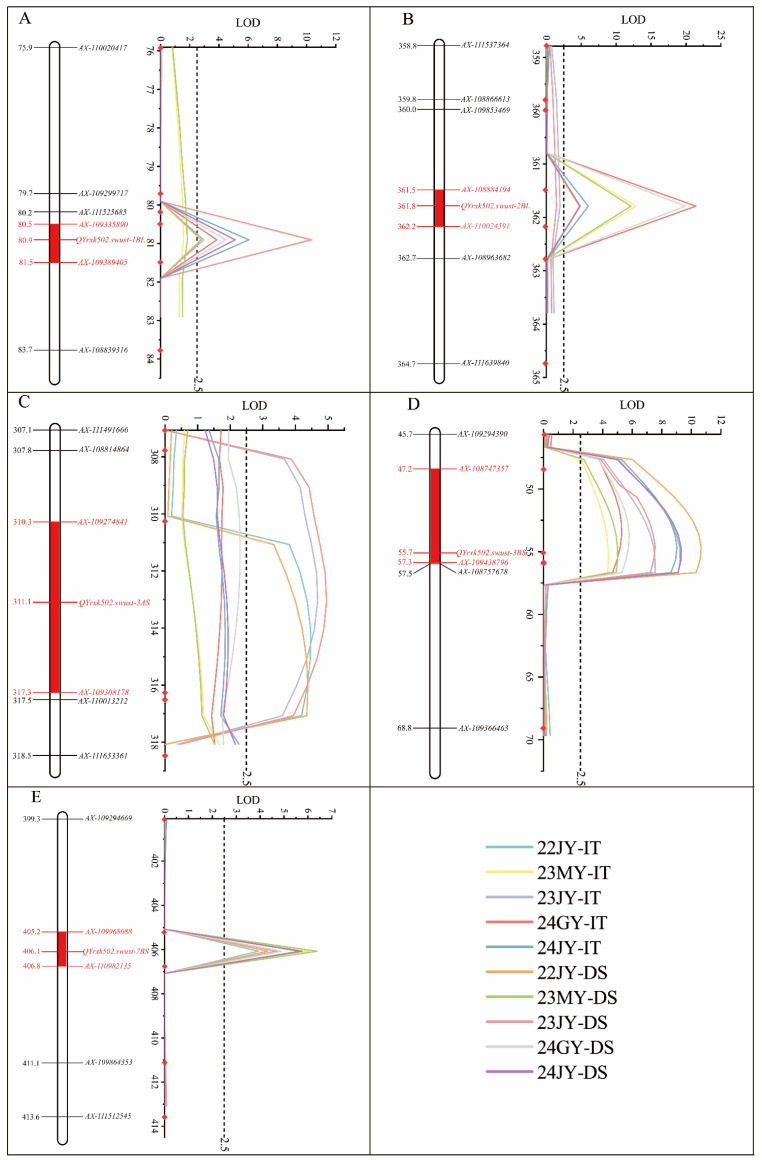
Quantitative trait loci (**A**–**E**) associated with stripe rust resistance on chromosomes 1BL, 2BL, 3AS, 3BS, and 7BS were mapped using infection type (IT) and disease severity (DS) data. The y-axis denotes genetic distance (cM), while the x-axis shows LOD values. The red rectangle on the genetic map indicates the location of each QTL.

**Figure 5 plants-13-02365-f005:**
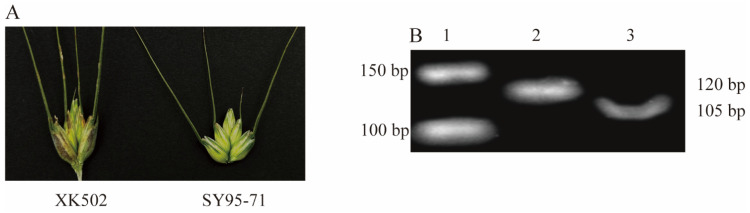
Parental XK502 and SY95-71 grain phenotypes (**A**); amplification of primer Xwms533 in XK12 and XK502 (**B**). ^1^ DNA maker; ^2^ XK12 (carrying the Yr30 gene) [32]; ^3^ XK502.

**Figure 6 plants-13-02365-f006:**
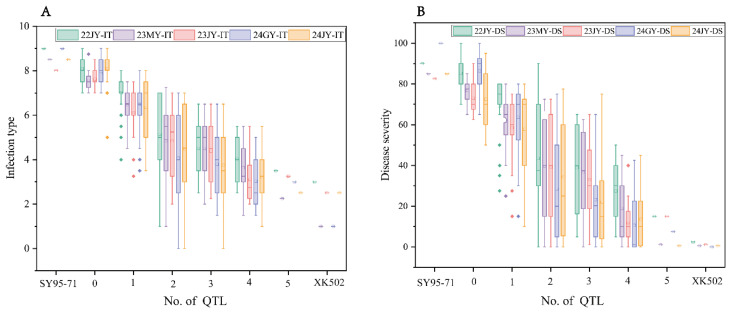
Effect of different infection type (**A**) and disease severity (**B**).

**Table 1 plants-13-02365-t001:** Correlation analysis between infection type (IT) and disease severity (DS) in recombinant inbred populations in five environments.

Environment	22JY	23MY	23JY	24GY	24JY
22JY	-				
23MY	0.57 (0.53) ^1^	-			
23JY	0.62 (0.73)	0.49 (0.61)	-		
24GY	0.58 (0.54)	0.68 (0.64)	0.63 (0.64)	-	
24JY	0.64 (0.65)	0.58 (0.62)	0.75 (0.73)	0.74 (0.70)	-

^1^ R-values for infection type (IT) and disease severity (DS), all R-values were highly significant, *p* < 0.001.

**Table 2 plants-13-02365-t002:** Analysis of variance for infection type (IT) and disease severity (DS) in a population of recombinant inbred lines.

Source of Variation	Infection Type				Disease Severity			
	df	Sum of Squares	Mean Square	F Value	df	Sum of Squares	Mean Square	F Value
Genotype	220	6903.90	31.38	30.92 ***	220	1,109,579.25	5043.54	34.53 ***
Environment	4	1388.81	347.20	342.09 ***	4	24,130.66	6032.66	41.3 ***
Genotype × Environment	880	2919.35	3.32	3.27 ***	880	528,512.63	600.58	4.11 ***
Error	1079	1096.16	1.02		1079	57,605.75	146.07	
h^2^_b_ ^1^	0.90				0.89			

*** The difference is highly significant at the *p* < 0.001 level; ^1^ h^2^_b_ broad-sense heritability.

**Table 3 plants-13-02365-t003:** Effects of different QTL on stripe rust infection type (IT) and disease severity (DS).

QTL	No. of RILs	IT	DS%
*QYrxk502*.*swust*-*1BL*	62	5.02	39.50
*QYrxk502*.*swust*-*2BL*	46	4.66	34.61
*QYrxk502*.*swust*-*3AS*	42	4.91	37.27
*QYrxk502*.*swust*-*3BS*	50	4.46	30.80
*QYrxk502*.*swust*-*7BS*	52	4.87	36.81

**Table 4 plants-13-02365-t004:** Correlation analysis between parent and RIL populations stripe rust phenotypes and agronomic traits.

Trait Name	IT	DS	PH	PTN	SL	TKW	GL	GW	LWR
IT	-								
DS	0.99 ***	-							
PH	−0.25 ***	−0.27 ***	-						
PTN	−0.34 ***	−0.35 ***	0.26 ***	-					
SL	−0.24 ***	−0.23 ***	0.47 ***	0.073 ^ns^	-				
TKW	−0.41 ***	−0.42 ***	0.45 ***	0.19 **	0.29 ***	-			
GL	−0.35 ***	−0.35 ***	0.40 ***	0.20 **	0.49 ***	0.75 ***	-		
GW	−0.32 ***	−0.33 ***	0.36 ***	0.15 *	0.16 *	0.88 ***	0.52 ***	-	
LWR	−0.11 ^ns^	−0.10 ^ns^	0.12 ^ns^	0.07 ^ns^	0.37 ***	0.05 ^ns^	0.60 ***	−0.34 ***	-

*** Indicates *p* < 0.001, the correlation is extremely significant; ** indicates *p* < 0.01, the correlation is extremely significant; * indicates *p* < 0.05, significant correlation; ^ns^ indicates *p* > 0.05, no correlation.

## Data Availability

Data are contained within the article.

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
