# Peer review of "Identification and Mapping of QTLs for Adult Plant Resistance in Wheat Line XK502"

_plants, 2024, doi:10.3390/plants13172365_

Round 1

Reviewer 1 Report

Comments and Suggestions for Authors

The study was focused on genome-wide QTL analysis for stripe rust resistance in wheat line XK502. According to the Authors, in three seasons (2022 to 2024), the stripe rust infection type (IT) and disease severity (DS) of parents and adult RILs under five environments were identified. A genetic linkage map was constructed using 12,577 single nucleotide polymorphism (SNP) markers. The Authors revealed five stripe rust resistant genes: QYrxk502.swust-1BL, QYrxk502.swust-2BL, QYrxk502.swust-3AS, QYrxk502.swust-3BS, and QYrxk502.swust-7BS. In addition, 16 lines were selected based on field stripe rust resistance and agronomic traits for further breeding and promotion of new varieties.

The obtained results are quite important in the research topic. The study is well organized. However, I have formulated few minor revisions:

-            I highly recommend increasing the graphical resolution of Figures 3-5 as well as Figure 6B.

-            I recommend removing Tables 4 and 6 into the Supplementary File.

-            Discussion part of the manuscript should be considerably improved by adding some new citations in the research topic.

-            The conclusions have to be re-written. At present, the most majority of the conclusions are simple repetition of the results.

-            Moderate editing of English language required.

Comments on the Quality of English Language

 Moderate editing of English language required.

Author Response

First of all, I would like to thank you very much for your comments on my manuscript, which have been very helpful in revising the manuscript. My feedback on your comments is as follows:

Comments 1: I highly recommend increasing the graphical resolution of Figures 3-5 as well as Figure 6B.

Response 1: Image resolution has been increased to 600.

Comments 2: I recommend removing Tables 4 and 6 into the Supplementary File.

Response 2: Tables 4 and 6 have been moved to the supplementary document.

Comments 3: Discussion part of the manuscript should be considerably improved by adding some new citations in the research topic.

Response 3: New references have been added to the Discussion section

Comments 4: The conclusions have to be re-written. At present, the most majority of the conclusions are simple repetition of the results.

Response 4: Added Outlook Changed to This study successfully integrated high-throughput 55K SNP sequencing with stripe rust phenotypes from both parental lines and 221 mapping populations, leading to the identification of five adult plant resistance QTLs. Each of these QTLs offers varying levels of resistance to stripe rust, highlighting their potential utility in crop improvement. While the associations between QYrxk502.swust-1BL and QYrxk502.swust-2BL and other QTLs require further validation, QYrxk502.swust-3AS, QYrxk502.swust-3BS, and QYrxk502.swust-7BS appear to represent novel contributions to our understanding of stripe rust resistance. Notably, the QTLs located on chromosomes 1BL, 2BL, 3BS, and 7BS exhibited stability across conditions, suggesting their reliability for breeding purposes. This study underscores the correlation between the clustering of resistance QTLs and enhanced resistance levels, indicating that selecting lines with multiple QTLs could yield more resilient cultivars. Furthermore, the identification of 16 promising lines from the 221 RILs through a dual approach of resistance and agronomic trait evaluation indicates a pathway toward the development of commercially viable wheat cultivars. Future efforts can focus on converting the SNP markers linked to these stable major effect QTLs into KASP or SSR markers to facilitate the breeding of wheat cultivars that possess lasting resistance to stripe rust.

Comments 5: Moderate editing of English language required.

Response 5: Already English processing has been done using MDPI's author services

Finally, we would like to thank you again for your attention and support, and hope that you will continue to follow our research work and offer more valuable comments and suggestions.

Reviewer 2 Report

Comments and Suggestions for Authors

Overall, I think this is quite interesting, and if the materials are available to the world community, this may be of considerable interest to a wider audience. However, at present I see numerous problems with the manuscript.

A general comment: after decades of QTL mapping it is difficult to publish unverified results. Here, the authors try to address the issue, by grouping their RILs according to the QTL content. This is a bit awkward, as it lumps together different loci, with different effects. Wouldn’t it make more sense to group the lines according to the actual QTL contents, averaging the reactions, to see the contribution of each one???? It certainly would make much more sense to me than grouping by the QTL numbers.

Second: you did not identify any genes. Some your QTLs are in regions known to harbor Yr genes, so either you detected those genes, or their alleles. To decide which is true you would have to isolate lines with just those alleles present, and challenge them with proper races. Differences in reactions would tell the story. As to new locations (are they really new, if hundreds of QTLs have already been identified?) you can only claim to have detected a QTL. Until you identify a possible candidate gene, and test it, you cannot speak of “genes”.

Third: English needs serious work. I give you an example, by re-working the abstract. Many f your sentences ought to be flipped, you keep repeating the same mistakes (resistant vs resistance), and quite a few others. Unfortunately, I have too much to do to try to re-write the thing.

 Why that "genome wide" in the title? It is a simple QTL analysis in a biparental population.

I do not like the first paragraph of Intro: to sweeping, too many generalizations, too dramatic. 

For one, stripe rust, while a serious disease, is a factor limiting production only in some areas, and when present.  An “epidemic” is a consequence of infection.

Use “cultivar”, not “variety”. “Variety” is a botanical term used in systematics.

Note the difference between “resistance” and “resistant”

In general, Intro strikes me as somewhat disorganized. It can be arranged much better, to flow from one point to another. For one, perhaps it would make sense to start the argument that identification and deployment of new genes for resistance is an ongoing struggle. In this line, a newly developed wheat line with good field level resistance was dissected using a mapping population……… and SNPs

Results: line 85-86. Are you saying that the resistant parent was highly susceptible?

Lines 89-90: the coordinates ought to be in M&M, no point in repeating here

Line 93: why not : highly susceptible reaction (IT = 8,9) and highly resistant reaction (IT =

Line 128: 14,164 markers were homozygous and polymorphism between the parents. Why not: 14164 markers were polymorphic between the parents?

Lines 13-132: rather odd! You must have had very many mis-scored markers. Given the average number of crossovers per chromosome of wheat (around 3), your maps should not exceed ca. 150 cM. 978 cM implies the average of 19.5 crossovers! You need to clean up your data and repeat the analysis.

My version of Abstract:

Stripe rust is a serious wheat disease occurring worldwide. At present, the most effective way to control it is to grow resistant cultivars. In this study, a population of 221 recombinant inbred lines (RILs) derived by single seed descent from a hybrid of a susceptible wheat line SY95-71 and a resistant line XK502 was tested in three crop seasons from 2022 to 2024 in five environments.  A genetic linkage map was constructed using 12,577 single nucleotide polymorphisms (SNP). Based on the phenotypic data of infection severity and the linkage map, five quantitative trait loci (QTL) for adult plant resistance (APR) were located using the inclusive composite interval mapping method. These five loci are: QYrxk502.swust-1BL, QYrxk502.swust-2BL, QYrxk502.swust-3AS, QYrxk502.swust-3BS, and QYrxk502.swust-7BS, explaining 5.67%-19.64%, 9.63%-36.74%, 9.58%-11.30%, 9.76%-23.98%, and 8.02%-12.41% of the phenotypic variation, respectively. All these QTLs originated from the resistant parent XK502. By comparison with the locations of known stripe rust resistance genes, three of the detected QTLs, QYrxk502.swust-3AS, QYrxk502.swust-3BS and QYrxk502.swust-7BS may harbor new, so far unidentified, genes. From among the tested RILs, 16 lines were selected with good field stripe rust resistance and acceptable agronomic traits for inclusion in breeding programs.

Comments on the Quality of English Language

English must be improved.

Author Response

First of all, I would like to thank you very much for your comments on my manuscript, which have been very helpful in revising the manuscript. My feedback on your comments is as follows:

Comments 1: A general comment: after decades of QTL mapping it is difficult to publish unverified results. Here, the authors try to address the issue, by grouping their RILs according to the QTL content. This is a bit awkward, as it lumps together different loci, with different effects. Wouldn’t it make more sense to group the lines according to the actual QTL contents, averaging the reactions, to see the contribution of each one???? It certainly would make much more sense to me than grouping by the QTL numbers.

Response 1: The grouping of rows according to the actual QTL content was added, and the effects of each QTL were calculated as QYrxk502.swust-3BS > QYrxk502.swust-2BL > QYrxk502.swust-7BS > QYrxk502.swust-3AS > QYrxk502.swust-1BL.

QTL

No. of RILs

IT

DS%

QYrxk502.swust-1BL

62

5.02

39.50

QYrxk502.swust-2BL

46

4.66

34.61

QYrxk502.swust-3AS

42

4.91

37.27

QYrxk502.swust-3BS

50

4.46

30.80

QYrxk502.swust-7BS

52

4.87

36.81

Comments 2: Second: you did not identify any genes. Some your QTLs are in regions known to harbor Yr genes, so either you detected those genes, or their alleles. To decide which is true you would have to isolate lines with just those alleles present, and challenge them with proper races. Differences in reactions would tell the story. As to new locations (are they really new, if hundreds of QTLs have already been identified?) you can only claim to have detected a QTL. Until you identify a possible candidate gene, and test it, you cannot speak of “genes”.

Response 2: Due to limited conditions, we used phenotypic comparison and marker detection methods for verification. By comparing the physical locations, we found that the physical intervals of QYrxk502.swust-1BL and QYrxk502.swust-3BS overlapped with Yr29 and Yr30, respectively. However, Yr29 is closely related to the leaf tip necrosis (LTN) gene (leaf tip necrosis occurs when the gene is carried), but no leaf tip necrosis was observed in XK502 in this study, so QYrxk502.swust-1BL is not Yr29. Yr30 is related to the morphological marker pseudoblack bark (PBC), and wheat carrying this gene will have a gradual darkening of the glumes and internodes at the late grain filling stage. gradually turn black. Although the disease-resistant parent XK502 in this study showed blackening of glumes in the late grain filling stage (Fig. 5A), we detected the flanking marker WMS533 for Yr30, which was not found in XK502 (Figure 5B), and thus QYrxk502.swust-3BS was not Yr30 either. This part was previously placed in the Discussion section and is now placed in the Conclusion. Relationships to reported QTLs are located in the Discussion section. It has been corrected to say that a QTL was detected.

Comments 3: Why that "genome wide" in the title? It is a simple QTL analysis in a biparental population.

Response 3: Already corrected to Identification and mapping of QTL for adult plant resistance in wheat line XK502.

Comments 4: I do not like the first paragraph of Intro: to sweeping, too many generalizations, too dramatic.

Response 4: Already accurate. Located in lines 26-35 of the article.

Comments 5: Use “cultivar”, not “variety”. “Variety” is a botanical term used in systematics.

Response 5: Already corrected to cultivar.

Comments 6: Note the difference between “resistance” and “resistant”

Response 6: Already corrected.

Comments 7: In general, Intro strikes me as somewhat disorganized. It can be arranged much better, to flow from one point to another. For one, perhaps it would make sense to start the argument that identification and deployment of new genes for resistance is an ongoing struggle. In this line, a newly developed wheat line with good field level resistance was dissected using a mapping population……… and SNPs

Response 7: Already corrected. It starts with the damage of stripe rust, then writes about the stripe rust genes that have been reported so far and their applications, then introduces the role of single nucleotide polymorphisms (SNPs) in QTL detection, and finally explains the purpose of this study. Located in lines 26-60 of the article.

Comments 8: Results: line 85-86. Are you saying that the resistant parent was highly susceptible?

Response 8: Lines 85–86 are the results of seedling identification, and because the parents were highly susceptible at the seedling stage, the QTL detected was for resistance in adult plants.

Comments 9: Lines 89-90: the coordinates ought to be in M&M, no point in repeating here

Response 9: Already deleted.

 Comments 10: Line 93: why not : highly susceptible reaction (IT = 8,9) and highly resistant reaction (IT =

Response 10: Already corrected to IT=8,9.

Comments 11: Line 128: 14,164 markers were homozygous and polymorphism between the parents. Why not: 14164 markers were polymorphic between the parents?

Response 11: Already corrected to 14164 markers were polymorphic between the parents

Comments 12: Lines 13-132: rather odd! You must have had very many mis-scored markers. Given the average number of crossovers per chromosome of wheat (around 3), your maps should not exceed ca. 150 cM. 978 cM implies the average of 19.5 crossovers! You need to clean up your data and repeat the analysis.

Response 12: The result was still the same after recalculation in the software QTL IciMapping4.2. The operation was as follows: 53,063 original markers were obtained from the 55K SNP chip of wheat after scanning the parents and the population, 14,164 markers were polymorphic between parents, and 12,577 markers were still available after removing the markers with deletion rate greater than 20% by BIN tool in the software QTL IciMapping4.2. The output map file of BIN tool was used as the input file to construct the genetic map, and the LOD Score, Recombination Frequency, Pairwise Distance were calculated by Kosambi mapping function to construct the genetic map. And we've made the 2.2 section more concise and streamlined by incorporating it into the 2.3 section.

Comments 13: My version of Abstract:

Response 13: Thank you very much for providing me with the summary version, which has been used in the article.

Comments 14: English must be improved.

Response 14: Already English processing has been done using MDPI's author services

Finally, we would like to thank you again for your attention and support, and hope that you will continue to follow our research work and offer more valuable comments and suggestions.

Round 2

Reviewer 2 Report

Comments and Suggestions for Authors

Nice job improving the manuscript. I appreciate you taking the suggestions seriously. 

I still think that grouping RILs by QTLs present would be more interesting than grouping by numbers. What you are doing is called "pyramiding", an age old concept. Hence, your conclusion sounds a bit naive (in the sense that what you observe is exactly what is expected). 

No matter what filters you used, you clearly have a bunch of misscored markers. This is why your maps are so long. Keep in mind: each misscored marker adds two crossovers to the total. Yes, the outcome may be the same (actually, similar) but my guess is, it would be more precise with misscored markers removed, or corrected. Generally, filters will not do it, it has to be done by hand, a rather tedious job.